# The Clinical Profile of Pediatric *M. pneumoniae* Infections in the Context of a New Post-Pandemic Wave

**DOI:** 10.3390/microorganisms13051152

**Published:** 2025-05-17

**Authors:** Mădălina Maria Merișescu, Gheorghiță Jugulete, Irina Dijmărescu, Anca Oana Dragomirescu, Larisa Mirela Răduț

**Affiliations:** 1Faculty of Dentistry, Department of Infectious Diseases, “Carol Davila” University of Medicine and Pharmacy, 050474 Bucharest, Romania; madalina.merisescu@umfcd.ro; 2National Institute for Infectious Diseases “Prof. Dr. Matei Bals”, European HIV/AIDS and Infectious Diseases Academy, No. 1 Dr. Calistrat Grozovici Street, 021105 Bucharest, Romania; radutlarisa@gmail.com; 3Faculty of General Medicine, Department of Paediatrics, “Carol Davila” University of Medicine and Pharmacy, 050474 Bucharest, Romania; irina.dijmarescu@umfcd.ro; 4Grigore Alexandrescu Children’s Emergency Clinical Hospital, 011743 Bucharest, Romania; 5Faculty of Dentistry, Department of Orthodontics and Dentofacial Orthopaedics, “Carol Davila” University of Medicine and Pharmacy, 050474 Bucharest, Romania; anca.dragomirescu@umfcd.ro

**Keywords:** *Mycoplasma pneumoniae*, lung damage, extrapulmonary manifestation, coinfection, Romania

## Abstract

*Mycoplasma pneumoniae* is an atypical bacterium with a tropism for the respiratory tract, but it can also cause numerous extrapulmonary involvements. The incidence of high rates varies in epidemiological waves, occurring at a frequency of 3–7 years. Since the end of 2023, an increase in the incidence of *M. pneumoniae* infection cases has been noted internationally. We conducted a retrospective study of children hospitalized and confirmed with *M. pneumoniae* infection in our clinic during the last two epidemiological peaks. We retrieved data from the hospital database and divided the patients into two groups, corresponding to the years 2018–2019 and 2023–2024, respectively. Fisher’s exact test was used to compare the proportions. In the years 2023–2024, we observed a higher incidence of patients with respiratory failure (*p* = 0.032), pleural reaction (*p* = 0.016), and pulmonary consolidation (*p* = 0.016) compared to the group in the years 2018–2019. Gastrointestinal involvement was more frequent in the years 2018–2019 (*p* = 0.004). The incidence of other extrapulmonary complications did not show significant differences. Infection with *M. pneumoniae* has varied clinical manifestations. In patients with community-acquired pneumonia, even in cases of consolidation, the possibility of infection with *M. pneumoniae* must also be considered.

## 1. Introduction

The bacterial infection caused by *Mycoplasma pneumoniae* (*M. Pneumoniae*) is primarily a respiratory tract infection, but it can also manifest in extrapulmonary sites [1].

The infection does not exhibit specific seasonality and occurs in epidemiological peaks every few years (3–7 years), during which it can be responsible for 4–8% of all cases of community-acquired pneumonia [1,2]. It primarily affects school-aged children and adolescents, with an incubation period of approximately 1–3 weeks [3].

*M. pneumoniae* is an atypical bacterium characterized by slow growth and the absence of a cell wall, requiring close contact for transmission [4]. The microorganism can interact with the host’s immune system and has been recently associated with the development of chronic conditions, such as asthma [5]. However, many infected individuals are asymptomatic, have self-limiting clinical symptoms, or experience mild upper respiratory tract involvement, which is why it is also known as “walking pneumonia’’ [2,5]. Typically, patients’ symptoms begin with headache, rhinorrhea, fatigue, or possibly myalgia. The flu-like symptoms at the onset of *M. pneumoniae* infection are characteristic of this bacterium. Another important feature is that the patient’s symptoms are more suggestive than the clinical examination, which is often normal [6]. A dry, irritating cough is usually the most common respiratory manifestation. The cough can be very frequent and may lead to chest pain or a prolonged refractory cough due to impaired ciliary function. It causes pneumonia, requiring hospitalization in approximately 5–10% of cases [2,6].

Atypically, *M. pneumoniae* can be associated with a fulminant course. There are patients whose onset is marked by neurological involvement, gastrointestinal symptoms, or hemolytic anemia. The most frequent neurological manifestations are encephalitis and meningitis, but there has also been a reported case of a 6-year-old boy who presented with facial paralysis [7].

There are differences in clinical manifestations depending on age groups. Children have a higher risk of severe infections, a longer duration of illness, and a higher frequency of coinfections compared to adults [8]. Age under 6 years is an independent risk factor for severe pneumonia [9].

According to the latest studies, hemolytic anemia is not exceptional even in adults, who initially have a severe course with a higher risk of thromboembolic events but ultimately a favorable prognosis [10]

Radiographic findings typically include unilateral or bilateral lower lobe or perihilar reticulonodular opacities, as well as bronchial cuffing. Patchy infiltrates, areas of consolidation, and linear atelectasis may also be observed [11]. Parapneumonic pleurisy occurs in 15% to 20% of patients who present with pneumonia [6].

Doxycycline and macrolides are first-line medications for the treatment of *M. pneumoniae* infection. Macrolide resistance has been reported, but most experts believe that macrolides remain the antibiotics of choice for treating *M. pneumoniae* infections in adults and children [12]. Tetracyclines or fluoroquinolones are alternative treatments in cases of macrolide resistance. Corticosteroids can be used in fulminant pneumonia caused by *M. pneumoniae*, as they have a beneficial effect on the exaggerated cell-mediated immune response. Several studies have demonstrated clinical efficacy in cases of macrolide-resistant *M. pneumoniae* pneumonia that did not require a change in antibiotic therapy [13].

The number of *M. pneumoniae* infection cases has increased globally since the second half of 2023, a trend that continued throughout 2024. In some European countries, the incidence was even higher than it had been during the pre-pandemic period. During the pandemic period, in line with the global situation, the number of cases decreased significantly [1]. A reduction in cases was also observed for other microorganisms; however, *M. pneumoniae* infection experienced reductions or even an absence of cases over a longer period [14].

In Romania, an increase in *M. pneumoniae* cases has been noted nationwide according to epidemiological surveillance bulletins [15]. However, there are no detailed data on the number of *M. pneumoniae* cases nor any articles on this topic published in the last 2 years.

During the pandemic, measures such as mask wearing, physical distancing, hand hygiene, and travel restrictions led to a significant limitation in the transmission of all respiratory infections [16]. The lack of exposure to viruses resulted in a decrease in collective immunity, so an increase in both the incidence and severity of respiratory infections was observed after the relaxation of restrictions [17]. Long COVID can also cause persistent inflammation in the body, affecting the normal functioning of the immune system and increasing susceptibility to other infections. Patients who have lung damage caused by repeated SARS-CoV-2 infections may have a reduced capacity to respond effectively to subsequent respiratory infections and are predisposed to more severe forms of illness [18].

The objective of this study was to describe the characteristics of pediatric patients hospitalized and diagnosed with *M. pneumoniae* during the last two epidemiological peaks, specifically the periods of 2018–2019 and 2023–2024. Additionally, the objective of this study was to provide an overview of the number of hospitalized cases with *M. pneumoniae* over the past 20 years.

## 2. Materials and Methods

### 2.1. Study Type and Area

This retrospective descriptive study was conducted in Southeastern Europe, north of the Balkan Peninsula, and it included cases of *Mycoplasma pneumoniae* infection in a tertiary hospital in Bucharest, the capital of Romania.

### 2.2. Study Population and Design

Participants in the current study were selected from the pediatric wards of the National Institute for Infectious Diseases “Prof. Dr. Matei Balș” in the periods of 2018–2019 and 2023–2024, respectively.

We created two groups representing the last two waves of *M. pneumoniae* infection, aiming to present the clinical and paraclinical characteristics of patients diagnosed with *M. pneumoniae* infection who required hospitalization. During the SARS-CoV-2 pandemic, specifically in the years 2020–2022, there was a very small number of hospitalized cases with *Mycoplasma pneumoniae* infection. Considering our goal to compare disease forms from epidemiological peaks, we excluded patients from years when the frequency of infection was low.

The patients studied were children aged 0–18 years and were divided into two groups based on the period of hospitalization, with the first group from 2018 to 2019 and the second group from 2023 to 2024. The positive diagnosis was established on epidemiological, clinical, laboratory, and paraclinical criteria. The laboratory samples analyzed included nasopharyngeal swabs and sputum for pathogen identification by PCR testing, as well as blood samples for serological testing. IgM antibody detection was used as a diagnostic method in 65% of the patients, and PCR testing was performed in 35% of the cases. We also analyzed demographic data (age, gender, and residence), clinical characteristics, coinfections, and complications of *M. pneumoniae* infection, as obtained from the patients’ medical records.

### 2.3. Statistical Analysis

The patients’ personal data were used after obtaining informed consent. Fisher’s exact test and Mann–Whitney U were used to compare proportions. Categorical variables are presented as numbers and percentages, while continuous variables are presented as the median and interquartile range (IQR). R software, version 4.4.2, was used (copyright (C) 2024 The R Foundation for Statistical Computing, R Core Team (2024). R: A language and environment for statistical computing. R Foundation for Statistical Computing, Vienna, Austria. URL: https://www.R-project.org (accessed on 2 April 2025)). The data from the period 2004–2024 were obtained from the hospital’s statistical program, Info World. We selected the period and the diagnosis of *M. pneumoniae* infection, both as primary and secondary diagnoses, and then counted the patients.

## 3. Results

### 3.1. Epidemiological and Demographic Data

Over the last two decades, at the National Institute of Infectious Diseases “Prof. Dr. Matei Balș”, the number of cases recorded annually has varied, showing both periods of rapid increase and periods of decline. During 2004–2014, the number of cases fluctuated but without extreme variations. Between 2014 and 2015, a large number of cases was recorded, with 64 cases identified. However, due to insufficient data, this group was excluded from the analysis. The lowest incidence was recorded during the SARS-CoV-2 pandemic between March 2020 and December 2022 (Figure 1).

During 2018–2019, we identified 61 cases, and in 2023–2024, 60 cases were recorded, categorized into two groups-Group 1 and Group 2. We performed a comparative analysis of these two groups of pediatric patients (Table 1).

No differences in incidence were identified according to gender; in both groups, we observed approximately equal proportions (female = 30 in Group 1 and Group 2, *p* = 1; male = 31 in Group 1 and 30 in Group 2, *p* = 1).

In terms of patient age, it is observed that the number of cases in both groups is very low in infants (0–1 year) and adolescents (13–18 years). Most cases were recorded in the age groups of 1–5 years and 6–12 years, respectively, in similar proportions for both groups. In Group 1, we identified nearly equal incidences in the 1–5-year-old age group (n = 29, 48%) and the 6–12-year-old age group (n = 27, 44%), with 8% (n = 5) in the 13–18-year-old age group. In Group 2, the highest incidence was observed in the 6–12-year-old age group (n = 30, 50%), followed by the 1–5-year-old age group (n = 23, 38%) and the 13–18-year-old age group (n = 6, 10%). The proportion of age group distribution was similar in both groups (*p* > 0.05). (Table 1).

Regarding the children’s residence in the two groups, no significant differences were observed between urban area patients (Group 1 = 38, Group 2 = 32) and rural area patients (Group 1 = 23, Group 2 = 28), *p* = 0.360.

### 3.2. Symptomatology and Paraclinical Aspects

The symptomatology of patients upon admission is primarily characterized by fever, cough, dyspnea, digestive symptoms (abdominal pain, vomiting), neurological symptoms (headache, paresthesia), and cutaneous and mucous membrane symptoms (rash, stomatitis). Fever was the most frequent symptom (Group 1: 89%, Group 2: 83%, *p* > 0.05), followed by cough (Group 1: 70%, Group 2: 88%, *p* = 0.023) and gastrointestinal symptoms (Group 1: 41%, Group 2: 16.6%, *p* = 0.004). Neurological manifestations (Group 1: 10%, Group 2: 12%, *p* > 0.05) and skin rash (Group 1: 11%, Group 2: 3%, *p* > 0.05) are the less common symptoms in *M. pneumoniae* infection in children in both groups (Table 2).

Clinical examination was not associated with a pneumonia diagnosis in all patients. In more than half of the patients, infiltrates were identified on radiography but showed no chest auscultation in a clinical exam (Group 1: n = 36, 59%; Group 2: n = 32, 53%). Decreased breath sounds were observed in one patient in the first group and in three patients in the second group. The patients presented more frequently with fine crackles than with wheezing. In the first group, 5% presented with wheezing, 31% presented with crackles, and one patient presented with both. In the second group, 45% presented with crackles, and 5% presented with both wheezing and crackles.

The average number of days from symptom onset was seven (IQR: 3–6.9) in the first group and nine (IQR: 5–10) in the second group.

An important characteristic of these patients is that despite most presenting with fever and a prolonged onset, the complete blood count showed normal leukocyte levels in 69% (n = 42) of Group 1 and 72% (n = 43) of Group 2 cases. Only 28% (n = 17) had leukocytosis in 2018–2019 compared to 25% (n = 15) in 2023–2024, with a *p* value greater than 0.05. C-reactive protein, fibrinogen, presepsin, and procalcitonin were analyzed as markers of inflammation [19,20,21]. In Group 1, the media CRP was 28 (IQR: 4–30.7), while in Group 2, it was 33 (IQR: 6–48) (Table 2).

In the first group, 15% of patients had unilateral or bilateral pulmonary consolidation compared to 32% in Group 2 (*p* = 0.016). In the first group, eight patients had no radiological abnormalities (*p* = 0.040) (Table 2).

The consolidation aspect does not correlate with leukocytosis or marked inflammatory syndrome in all cases. There were also children with interstitial disease who had marked leukocytosis, with a percentage of 6% in the first group and 8% in the second group.

Pulmonary involvement was more severe in Group 2, with pleural reaction in 10% and 13% presenting respiratory failure compared to the 2018–2019 group, where pleural reaction and respiratory failure were identified in only one patient (Table 2).

In Group 1, patients with pulmonary consolidation on X-rays were aged between 6 and 12 years (mean age 10.1; SD 1.68). The mean age of those with alveolar–interstitial involvement was 5.09 years (IQR 1.9–7; SD 4.14), while the mean age of patients with interstitial involvement was 5.8 years (SD 3.6). In Group 2, the mean age of children with consolidation was 8.31 (SD 3.77), followed by slightly lower ages for those with alveolar–interstitial and interstitial involvement, at 6.66 (SD 3.5) and 6.25 (SD 3.9), respectively (Figure 2).

### 3.3. Extrapulmonary Complications

Extrapulmonary complications of *M. pneumoniae* infection in children were gastrointestinal manifestations, neurological complications, skin manifestations, hematological manifestations such as hemolytic anemia, and ENT involvement such as acute otitis media and parotitis (Table 3).

Gastrointestinal complications were significantly higher in the first group (Table 3). We identified complications such as reactive hepatitis, with seven cases in Group 1 and five cases in Group 2, as well as gastrointestinal manifestations such as diarrhea and vomiting (12 and 15 cases in Group 1 and four and three cases in Group 2, respectively). No pancreatic involvement was identified in any patient.

In both groups, dermatological manifestations were identified, including urticarial rash, polymorphic erythema, and petechial rash (Group 1: n = 15; Group 2: n = 12; *p* > 0.05). However, in Group 1, severe dermatological damage, classified as Stevens–Johnson-like syndrome, was identified in four patients, whereas in Group 2, no cases with this manifestation were identified.

Neurological complications, including acute encephalitis (two cases in the first group) and meningitis (two cases in each group), as well as neurological symptoms persisting for more than 24 h, such as altered consciousness (including drowsiness in three cases of the first group) were observed.

We identified hemolytic anemia in only two patients in 2018 who required transfusions but with a favorable outcome.

### 3.4. Coinfections

Coinfections with one pathogen were identified in 19 children in Group 1 (31%) and in 18 children in Group 2 (30%). Multi-organism coinfections occurred in three cases (5%) in Group 1 and ten cases (17%) in Group 2. There are no significant major differences between the total number of coinfections between the two groups (Group 1: n = 22; Group 2: n = 28; *p* = 0.271), but we observed a higher number of those who had coinfections with more than one pathogen in Group 2 compared to Group 1, with *p* = 0.044 (Table 4).

These were identified by serology or by rapid tests and PCR from nasal, pharyngeal, and nasopharyngeal exudate. We identified viral coinfections, including RSV, adenovirus, SARS-CoV-2, and EBV, as well as coinfections with beta-hemolytic Streptococcus or Streptococcus pneumoniae.

### 3.5. Hospitalization Days

The average number of hospitalization days was higher in the first group (Group 1-N = 61, median 7, IQR 5–9; Group 2-N = 60, median 6, IQR 5–7, *p* < 0.001).

Even though there were more coinfections in Group 2, the length of hospital stay was longer in the first group (Group 1-N = 23, median = 7, IQR 5–8,5; Group 2-N = 28, median = 6, IQR 5–7; *p* < 0.001) (Table 5).

In 2018–2019, the median number of hospitalization days for patients without coinfections was 7.5 (N = 38, IQR 6–9), and for those with coinfections, it was seven (N = 23, IQR 5–8.5), representing a duration of hospitalization without statistically significant differences (*p* = 0.577). In 2023–2024, the median length of hospital stay was 6 days (N = 28, IQR 5–7) in children with coinfections and 5 days (N = 32, IQR 4.75–7) in those without coinfections (Table 5). Comparing the duration of hospitalization in Group 2, we observed that patients with coinfections had a longer duration of hospitalization (*p* = 0.009).

A longer length of hospital stay in 2018–2019 was also observed in patients with or without corticosteroid therapy and patients without comorbidities (*p* < 0.001).

No statistically significant differences were observed between the two groups regarding the length of hospital stay in patients with comorbidities (Group 1-N = 12, median = 7.5, IQR 6–8.5; Group 2-N = 25, median = 6, IQR 5–9; *p* = 0.447) (Table 5).

### 3.6. Comorbidities

The main comorbidities were recurrent wheezing, atopic dermatitis, immunodepression, nutritional status disorders, autoimmune diseases, and neurological disorders. In the first group, people with comorbidities had a 14% higher risk of developing complications compared to those without comorbidities (OR-1.14, Cl 95%-0.31–3.86). A total of 13 children had comorbidities, of which 8 had complications. Of those without comorbidities, 28 individuals experienced complications, while 20 did not (Figure 3).

In the second group, 33 people had complications (55%), of whom 17 had comorbidities and 16 did not. In the case of patients without complications, 8 had comorbidities and 19 did not. The risk of complications was 25% higher for patients with comorbidities (OR 2.5, Cl-0.85–7.2). A larger sample of patients would likely yield a clearer result regarding the association between comorbidities and complications and would also indicate whether there is a statistically significant difference (Figure 4).

### 3.7. Treatment

The antibiotic therapy administered was clarithromycin in 99.1% of cases. Only one patient received doxycycline because he had received macrolide treatment at home. It was probably not a resistance to medication but a low compliance with antibiotic administration. A total of 30% in the first group and 58% in the second group received, in addition to macrolide treatment, treatment with a third-generation cephalosporin. Vancomycin was associated with 7% of patients in the first group and 18% in the second group. This represents the patients with leukocytosis, biological inflammatory syndrome, or identified bacterial coinfections like Streptococcus pneumoniae. Additionally, empirical antibiotic therapy targeting Gram-positive bacteria was initiated in patients presenting with radiographic consolidation, pending identification of the etiological pathogen.

## 4. Discussion

The data observed in our study are consistent with data from Europe. The previous wave was observed in 2019 in several countries, followed by a dramatic decline during the pandemic [19]. The global decline in *Mycoplasma pneumoniae* infections can be attributed to the implementation of non-pharmaceutical interventions to curb the spread of SARS-CoV-2. These measures included enhanced hygiene practices, home isolation, community restrictions, and targeted testing. According to international data, *M. pneumoniae* infection has re-emerged since 2023 at a rate comparable to before the pandemic. An increased number of cases has been reported in Denmark, Sweden, the Netherlands, and Slovenia, as well as in Asia and Singapore [4,19].

A single elevated IgM antibody titer can be a diagnostic tool according to guidelines. IgM titers begin to rise 7–10 days after infection and peak after 3–4 weeks [22].

Studies in the literature have shown that preschoolers infected with *Mycoplasma pneumoniae* have a significantly higher risk of developing severe pneumonia due to an insufficiently developed immune system or a higher frequency of bacterial coinfections [23,24]. In our study, children who required hospitalization were in the 1–5-year-old age group (Group 1-48% and Group 2-38%) and in the 6–12-year-old age group (Group 1-44% and Group 2-50%). According to previous studies, preschoolers were more prone to severe forms of the disease and required hospitalization. The number of school-aged children who required hospitalization probably correlates with the higher prevalence of *M. pneumoniae* infection at this age.

In our study, consistent with data from the literature, patients with *M. pneumoniae* infection experience a prolonged onset and long-lasting fever [25]. Gastrointestinal symptoms are common in *M. pneumoniae* infection. Søndergaard et al. identified diarrhea and vomiting in 33% of patients in a 2-year study [26,27].

In contrast to studies from Denmark that described an increase in cases of mucocutaneous rashes compared to the pre-pandemic period [1], our research identified dermatological involvement in 15 patients between 2018 and 2019 and 12 patients between 2023 and 2024. We identified Stevens–Johnson-like syndrome in four patients in the first group. Among the organisms associated with Stevens–Johnson syndrome (SJS), *M. pneumoniae* is the most common. Latsch et al. described severe ulcerative stomatitis with conjunctivitis and genital erosions but without skin involvement as incomplete Stevens–Johnson syndrome [28]. The term RIME (reactive infectious mucocutaneous eruption) was later adopted. Compared to SJS, this condition has a milder course and generally good prognosis [21,27,29].

In Spain, a multicenter study has observed, since the end of 2023, a more severe evolution of pediatric patients with *M. pneumoniae* infection, leading to an increase in hospital admissions and even the need for PICU support [30]. Qianyue Wu et al. concluded that before the COVID-19 pandemic, the proportion of patients with severe pneumonia caused by *M. pneumoniae* was generally above 10%. Some studies have shown that the prevalence of severe pneumonia has exceeded 20% during and after the pandemic [31]. It is speculated that this may be related to immunological impairment caused by COVID-19 infection and increased resistance to macrolides [32,33]. Accordingly, our study identified a higher proportion of children with pleural reactions and those with respiratory failure in 2023–2024 compared to the pre-pandemic period. In contrast, the most recent description of cases in the United States concluded that they did not identify an increase in the severity of the disease. They did not identify a higher number of patients requiring hospitalization or mechanical ventilation [34].

The rate of coinfections among children with community-acquired pneumonia is high. Rates of 27% with viral coinfections are cited in a study from Denmark, 38.75% in a study from China, and 43.4% in a study from Korea [26,35,36]. In our study, the rate of coinfections ranged from 36% in the first group to 46% in the second group. This difference between the two groups can be explained by the existence of several diagnostic methods in 2023–2024 that helped identify coinfections or by the fact that post-pandemic, the rate of all respiratory tropism infections has increased [17,37,38].

Previous studies have demonstrated that *M. pneumoniae* has developed resistance to macrolides through the acquisition of point mutations. According to the CDC, China has reported the highest level of resistance, at approximately 80%, followed by Japan, with over 50%. In Europe, a percentage of approximately 5% has been reported, and in the United States, most studies have shown percentages of 10% [28,37,38]. In our study, patients responded favorably to macrolide treatment.

The addition of beta-lactam antibiotics is indicated in children over 4 years of age with suspected atypical pneumonia who fulfill the following criteria: patients present with chills, leukocytes > 15,000 microL, CRP > 35 to 60 mg/L, and a lack of response to outpatient therapy with macrolides [20,21,27,39].

The study’s main limitations include its retrospective nature, single-center setting, and the fact that only hospitalized cases were included in the analysis. This may overestimate the severity of *M. pneumoniae* infection cases because we do not have a comprehensive picture of both hospital and non-hospital cases.

## 5. Conclusions

In conclusion, our study is significant because it raises concerns about the resurgence of *M. pneumoniae* infections among children of all ages. Physicians must be vigilant and look for this bacterium as a potential etiology of community-acquired pneumonia, especially in patients who have received treatment with beta-lactam antibiotics and have experienced an unfavorable outcome. Clinical symptoms can be significantly varied. Most present with cough and fever, but with good general condition, leading to a longer time from onset to diagnosis. At the post-pandemic peak, an increase in cases presenting with pleural reaction and respiratory failure was observed; 32% of cases in the second group presented with consolidation compared to 15% in the first group. Even though patients in the second group presented with more significant pulmonary impairment, the average hospitalization period was even shorter than in the first group. Thus, the evolution was favorable under treatment, and we did not identify macrolide resistance in our study. In Group 2, a higher number of patients with multiple coinfections was identified (n = 10, *p* = 0.044). Complications occurred with a slightly higher predisposition in children with comorbidities (a 14% higher risk in the first group and 25% higher risk in the second group). No deaths or admissions to intensive care were recorded.

## Figures and Tables

**Figure 1 microorganisms-13-01152-f001:**
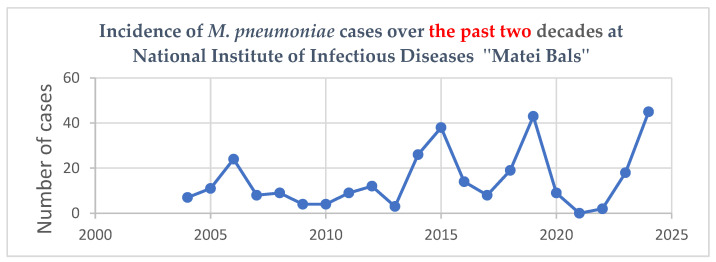
Incidence of M. pneumoniae cases in the last two decades at the National Institute of Infectious Diseases, “Prof. Dr. M. Bals”.

**Figure 2 microorganisms-13-01152-f002:**
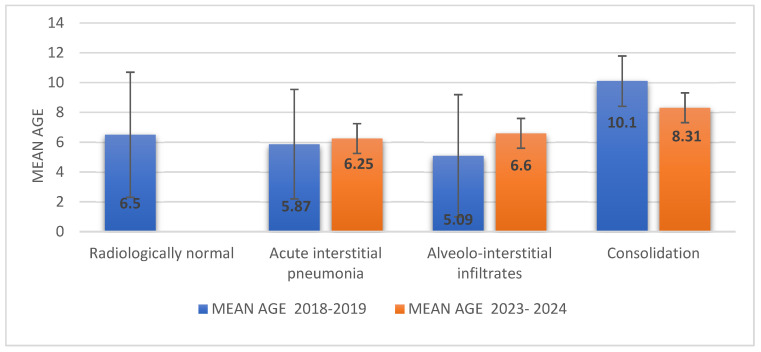
Comparative degree of lung damage by age group in both studied groups.

**Figure 3 microorganisms-13-01152-f003:**
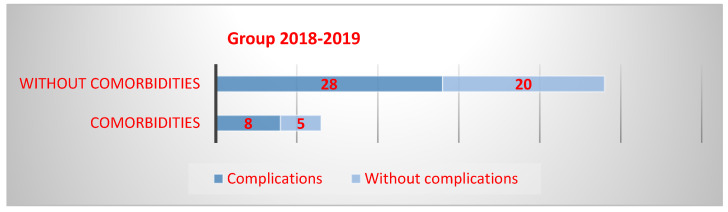
Occurrence of complications in patients with comorbidities in Group 1.

**Figure 4 microorganisms-13-01152-f004:**
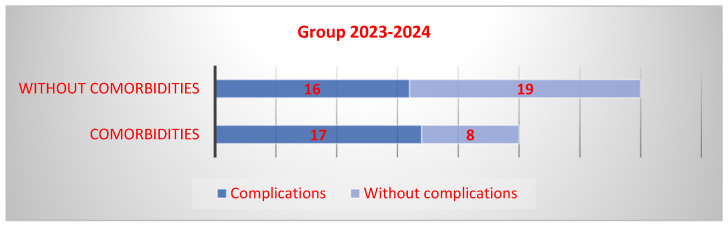
Occurrence of complications in patients with comorbidities in Group 2.

**Table 1 microorganisms-13-01152-t001:** Demographic characteristics of study participants. Significance (*p* < 0.05).

	Group 1 (2018–2019) (N = 61)	Group 2 (2023–2024) (N = 60)	*p* Value
**Gender, n (%)**			*p* = 1
Female	30 (49%)	30 (50%)	
Male	31 (51%)	30 (50%)	
**Age**			
<1 year old	0	1 (2%)	*p* = 0.496
1–5 years old	29 (48%)	23 (38%)	*p* = 0.360
6–12 years old	27 (44%)	30 (50%)	*p* = 0.587
13–18 years old	5 (8%)	6 (10%)	*p* = 0.762
**Residence**			*p* = 0.360
Urban	38	32	
Rural	23	28	

**Table 2 microorganisms-13-01152-t002:** Clinical and paraclinical characteristics of patients in both groups. Significance (*p* < 0.05).

	Group 1, 2018–2019 (N = 61)	Group 2, 2023–2024 (N = 60)	*p* Value
**Symptoms**			
Fever	54 (89%)	50 (83%)	*p* > 0.05
Cough	43 (70%)	53 (88%)	*p* = 0.023
Dyspnea	0	7 (11.6%)	*p* = 0.006
Abdominal pain, vomiting, diarrhea	25 (41%)	10 (16.6%)	*p* = 0.004
Neurological impairment (paresthesia, headache)	6 (10%)	7 (12%)	*p* > 0.05
Skin rash and mucositis	10 (16%)	7 (12%)	*p* > 0.05
**Paraclinical investigations**			
Normal leukocytes	42 (69%)	43 (72%)	*p* > 0.05
Leukocytosis	17 (28%)	15 (25%)	*p* > 0.05
Leucopenia	2 (3%)	2 (3%)	*p* > 0.05
C-reactive protein (mg/L), mean	28 (IQR 4–30.7)	33 (IQR 6–48)	
Fibrinogen, mean	381 (IQR 303–443)	380 (IQR 310–434)	
**Radiographic findings**			
Radiologically normal	8 (13%)	1 (2%)	*p* = 0.040
Acute interstitial pneumonia	24 (39%)	22 (37%)	*p* > 0.05
Alveolar–interstitial infiltrates	20 (31%)	18 (33%)	*p* > 0.05
Consolidation	9 (15%)	19 (32%)	*p* = 0.016
Pleural reaction	1	6 (10%)	*p* > 0.05

**Table 3 microorganisms-13-01152-t003:** Extrapulmonary complications of the study participants. Significance (*p* < 0.05).

Extrapulmonary Complications	Group 1	Group 2	*p*
Dermatological	15	12	*p* > 0.05
Neurologic	6	2	*p* > 0.05
Gastrointestinal	37	17	*p* < 0.001
Hemolytic anemia	2	0	*p* > 0.05
Otitis	5	2	*p* > 0.05
Parotitis	2	0	*p* > 0.05

**Table 4 microorganisms-13-01152-t004:** Coinfections of the study participants. Significance (*p* < 0.05).

	Group 1 (2018–2019) (N = 61)	Group 2 (2023–2024) (N = 60)	*p* Value
Coinfections	22 (36.07%)	28(46.67%)	0.271
Coinfections with one pathogen	19 (31.15%)	18 (30%)	1
Multi-organism coinfections	3 (4.92%)	10 (16.67%)	0.044

**Table 5 microorganisms-13-01152-t005:** Number of days of hospitalization of patients in the study groups. Significance (*p* < 0.05).

Hospitalization Days	Group 1 (2018–2019)	Group 2 (2023–2024)	*p* Value
All patients	n = 61 (Median = 7, IQR 5–9)	n = 60 (Median = 6, IQR 5–7)	*p* < 0.001
With coinfections	n = 23 (Median = 7, IQR 5–8.5)	n = 28 (Median = 6, IQR 5–7)	*p* < 0.001
Without coinfections	n = 38 (Median = 7.5, IQR 6–9)	n = 32 (Median = 5, IQR 4.75–7)	*p* < 0.001
With corticosteroid therapy	n = 28 (Median = 8, IQR 6–10)	n = 21 (Median = 6, IQR 5–7)	*p* < 0.001
Without corticosteroid therapy	n = 33 (Median = 6, IQR 5–8)	n = 39 (Median = 5, IQR 4.5–7)	*p* < 0.001
With comorbidities	n = 12 (Median = 7.5, IQR 6–8.5)	n = 25 (Median = 6, IQR 5–9)	*p* = 0.447
Without comorbidities	n = 49 (Median = 7, IQR 5–9)	n = 35 Median = 6 (IQR 5–7)	*p* < 0.001

## Data Availability

The datasets generated and analyzed during the current study are available from the corresponding author upon request.

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
