# Peer review of "The Clinical Profile of Pediatric M. pneumoniae Infections in the Context of a New Post-Pandemic Wave"

_microorganisms, 2025, doi:10.3390/microorganisms13051152_

Round 1
Reviewer 1 Report
Comments and Suggestions for Authors
All of the comments and suggestions are listed in detail in the attached PDF file.

Author Response
1 Reviewer’s Report The manuscript entitled “Mycoplasma Pneumoniae—A New Cycle with More Significant Lung Damage” is an interesting article which aspires to describe in detail the characteristics of pediatric patients hospitalized and diagnosed with M. pneumoniae during the last two epidemiological peaks, specifically the periods 2018-2019 and 2023-2024. Yet, the article has several limitations that were not addressed by the authors. Below are the main concerns raised during the revision: ➢ Major issues: The Methodology Section:
Point [1]: The methodology section does not offer sufficient data to allow study replication. ➢The readability of this section could be improved. Hence, it would be useful to have the methodology section restructured as follows:
- Start the section by adding a subtitle to indicate the study type and area, as follows: ➢ Study type and area: “This retrospective descriptive study was carried out in…. Add a brief description of the geographical location of the National Institute for Infectious Diseases.
The methodology section was changed and restructured, adding details about the geographical location of the institution where the children were investigated, as well as the patient inclusion criteria. Subtitles were added and all previous text was reformulated
- Add the following subtitle to introduce the study subjects: ➢ Study population and design: “Participants in the current study were selected from the the pediatric wards of the National Institute for Infectious Diseases” Prof. Dr. Matei BalÈ™” in the periods 2018-2019 and 2023-2024, respectively.
- This was done. We added a subtitle to introduce the study subjects.
- If there were any gaps in the analyzed data, I suggest stating the reasons in the methodology section.
- I explained that we did not consider it appropriate to include the patients from the pandemic period because there were very few of them, and we wanted to compare the characteristics of patients who were part of epidemiological waves
- Starting line 85, the authors describe the statistical analysis applied in the study. Therefore, consider adding a separate clear subtitle, named: Statistical analysis:
- This was done.
Point [2]: ➢ There is a critical shortage in the analyzed demographic data. The authors have only included two variables, which are age and sex, without defining the calculated P value. What about other important variables, like the residence, why was it not included among the tested parameters?
- This was done. We added p-values ​​for both sex and age. We also added the residence of the children in the study and calculated p for these data as well.
The (Results) section: Point [3]: Results are not clearly stated, and data are not presented properly.
- The authors start the results section by describing the features of Figure 1. Yet, in lines 94 to 96, the authors describe the data represented in Table 1 instead. ➢ Consider describing in a clear informative way the data shown in Figure 1.
The results were reformulated, the figures and tables were explained better. I started by describing the data in Figure 1 more clearly and informatively.
- Figure 1 shows the incidence of M. pneumoniae during the period from 2004 to 2024. Hence, it depicts the incidence of cases over the past two decades. ➢ Consider applying the previous correction to the title of the chart.
The chart title has been corrected and the change has been made.
- Table 1: The table critically lacks the calculated P value.
The P's were calculated and Table 1 was recreated
- Table 2 is poorly presented. The authors have repeated the same data in the same table (Page 3). Some headings are broken between the rows.
We reorganized the table. The data has been rearranged, in a clear and concise manner
- Table 2: Consider the following correction (Group 1 (2018-2019) instead of 2029.
There was a typo that escaped us, it has been corrected
Point [4]: In lines 182 to 221, the authors only mention their work's findings in a descriptive manner. The data were not presented as tables or graphs, nor were they statistically analysed.
- Additional tables were created to support the interpretation of the results, allowing for a clearer presentation and visualization of the data through figures and tabulated formats.
➢ Minor issues: The Introduction Section: Point [5]: The authors have highlighted the current global status of M. pneumoniae infection. Yet, what about the situation/prevalence of M. pneumoniae infection in Romania? Consider adding a few lines to inform the reader about the current status of M. pneumoniae infection among Romanian citizens.
In Romania, a nationwide increase in Mycoplasma pneumoniae infections has been observed, according to data from epidemiological surveillance bulletins. However, there is a lack of detailed information regarding the exact number of cases, regional distribution, severity of illness, or therapeutic interventions. Our official reporting platforms indicate a general rise in M. pneumoniae-associated pneumonia but are unable to provide region-specific data, the number of severe cases per area, or information on the treatments administered. Therefore, European studies were used as the primary reference for comparing and contextualizing the findings of our research.
Reviewer 2 Report
Comments and Suggestions for Authors
Dear Authors,
Mycoplasma pneumoniae is a major bacterial cause of respiratory tract infection. Several studies have demonstrated that the incidence of high rates of the infection fluctuates in waves, happening at a frequency of 3-7 years. In this interesting work, the authors compare the clinic characteristics and complementary results of pediatric patients hospitalized and diagnosed with M. pneumoniae infection during the last two epidemiological peaks (2018-2019 and 2023-2024). Between these two peaks, the COVID-19 pandemic took place. As the authors analyse, and in harmony with what has been described in other studies, this pandemic could have influenced the incidence and clinical characteristics of the peak after it. However, an absent piece of information that could have been part of the analyses is the possible presence of manifestations of Long COVID prior to M. pneumoniae infection in part of the patients at the peak of 2023-2024. I suggest that authors include, and discuss, that information in a possible second version of the manuscript.
In the section Discussion, between line 282 y 285, the authors declare some limitations of the work: “…its retrospective nature, single-center setting, and the fact that only hospitalized cases were included in the analysis”. Of them, I consider that the only inclusion of hospitalized cases is the more important limitation of this study given that, as the authors recognize, “This may overestimate the severity of M. pneumoniae infection cases because we do not have a comprehensive picture of both hospital and non-hospital cases”. Taken into account it, I suggest to the authors that in the title and in the texts of a possible second version of the manuscript they do not refer so categorically to 'A New Cycle with More Significant Lung Damage'.
Author Response
Mycoplasma pneumoniae is a major bacterial cause of respiratory tract infection. Several studies have demonstrated that the incidence of high rates of the infection fluctuates in waves, happening at a frequency of 3-7 years. In this interesting work, the authors compare the clinic characteristics and complementary results of pediatric patients hospitalized and diagnosed with M. pneumoniae infection during the last two epidemiological peaks (2018- 2019 and 2023-2024). Between these two peaks, the COVID-19 pandemic took place. As the authors analyse, and in harmony with what has been described in other studies, this pandemic could have influenced the incidence and clinical characteristics of the peak after it. However, an absent piece of information that could have been part of the analyses is the possible presence of manifestations of Long COVID prior to M. pneumoniae infection in part of the patients at the peak of 2023-2024. I suggest that authors include, and discuss, that information in a possible second version of the manuscript.
- Thank you for your insightful review and constructive suggestions. They are highly appreciated. Regarding the cohort of children in the 2023–2024 group, quantifying the presence of SARS-CoV-2 infection in their medical history proves to be challenging. Moreover, it is important to consider that Long Covid may also manifest in individuals who experienced asymptomatic or mild forms of the infection. In our study, it is hypothesized that these patients were more likely predisposed to severe forms of illness due to their exposure to various microorganisms following a prolonged period of isolation, rather than attributing the increased severity of the disease directly to Long Covid.
During the SARS-CoV-2 pandemic, stringent isolation measures effectively reduced the incidence of common respiratory infections, including Mycoplasma infections. However, our findings suggest that the post-pandemic cases of Mycoplasma were significantly more severe compared to pre-pandemic cases, which could potentially reflect the influence of prior SARS-CoV-2 exposure on the current presentation of infections. This shift in the severity of infections post-pandemic may therefore be indicative of the complex interplay between viral and bacterial infections, as well as the broader immunological consequences of the isolation measures implemented during the pandemic.
In the section Discussion, between line 282 y 285, the authors declare some limitations of the work: “…its retrospective nature, single-center setting, and the fact that only hospitalized cases were included in the analysis”. Of them, I consider that the only inclusion of hospitalized cases is the more important limitation of this study given that, as the authors recognize, “This may overestimate the severity of M. pneumoniae infection cases because we do not have a comprehensive picture of both hospital and non-hospital cases”. Taken into account it, I suggest to the authors that in the title and in the texts of a possible second version of the manuscript they do not refer so categorically to- A New Cycle with More Significant Lung Damage
The title indeed presents a generalization of considerable scope compared to the data obtained in our study. We considered that all cases with more severe outcomes were hospitalized, while those treated in outpatient settings by family doctors, who did not require hospitalization, were automatically categorized as mild cases. Consequently, our study involved a comparison between severe cases (requiring hospitalization) from the pre- and post-pandemic cohorts, leading to the conclusion that the most recent wave was associated with more severe pulmonary involvement. Therefore, the title will be revised and reformulated to better reflect the scope and findings of the study.
Clinical profile of pediatric M. pneumoniae infections in the context of a new post-pandemic wave
Reviewer 3 Report
Comments and Suggestions for Authors
The authors presented the characteristics of pediatric patients hospitalized with M. pneumoniae at the National Institute for Infectious Diseases "Prof. Dr. Matei BalÈ™" in the periods 2018-2019 and 2023-2024.
Some comments are listed below.
Line 36. Please provide a description of M. pneumoniae bacteria, as well as typical and atypical symptoms of infection with this bacterium.
In the Introduction section, please include cases of M. pneumoniae infection in children with typical and atypical clinical presentations. Are there differences in the course of disease between adults and children? The main treatment regimens for infection in children should also be included in the introduction.
Figure 1 shows data on registered cases of M. pneumoniae in the period 2004-2024. How was this data obtained? Please provide a description of this figure on line 93. Why were the 2018-2019 and 2023-2024 groups created for further study?
Line 223. Please discuss the consistency of your study with the European data.
Author Response
The authors presented the characteristics of pediatric patients hospitalized with M. pneumoniae at the National Institute for Infectious Diseases Prof. Dr. Matei BalÈ™, in the periods 2018-2019 and 2023-2024.
Some comments are listed below.
Line 36. Please provide a description of M. pneumoniae bacteria, as well as typical and atypical symptoms of infection with this bacterium. In the Introduction section, please include cases of M. pneumoniae infection in children with typical and atypical clinical presentations. Are there differences in the course of disease between adults and children? The main treatment regimens for infection in children should also be included in the introduction.
We have revised the introduction to include an updated classification of patients, distinguishing between typical and atypical manifestations of Mycoplasma pneumoniae infection. Patients presenting with flu-like symptoms and frequent cough were categorized as having typical clinical features. Additionally, both the literature and our study describe cases in which respiratory symptoms are not predominant; instead, patients initially present with gastrointestinal disturbances, hemolytic anemia, neurological or dermatological manifestations. These atypical features are more frequently observed in pediatric populations.
Although the clinical presentation does not differ significantly between adults and children, our study was conducted in a pediatric clinic, and therefore only children were included and monitored. Moreover, the therapeutic approaches employed were also introduced in the context of the current literature on the subject.
Figure 1 shows data on registered cases of M. pneumoniae in the period 2004-2024. How was this data obtained? Please provide a description of this figure on line 93. Why were the2018-2019 and 2023-2024 groups created for further study?
The data presented in Figure 1 were extracted using the Info World statistical software implemented in our hospital. We selected the time frame 2004–2024 and included patients with a primary diagnosis of Mycoplasma pneumoniae infection. The total number of cases was counted, and periods with the highest incidence were identified. All selected cases were then subjected to detailed evaluation, which included both individualized case analysis, focusing on specific clinical features, and a comprehensive statistical assessment.
Two study groups were established, corresponding to the most recent waves of M. pneumoniae infection recorded during 2018–2019 and 2023–2024, respectively. The objective was to compare the clinical and paraclinical characteristics of patients diagnosed before the COVID-19 pandemic with those diagnosed in the post-pandemic period. In 2020, nine hospitalized cases of M. pneumoniae infection were recorded, followed by none in 2021 and only two cases requiring hospitalization in 2022. Due to the low number of cases and in light of our objective to compare hospitalized patients (reflecting more severe forms of the disease) from the pre-SARS-CoV-2 period with those from the post-pandemic era, these intermediary cases were excluded from the comparative analysis.
Line 223. Please discuss the consistency of your study with the European data.
A comprehensive review of existing studies on Mycoplasma pneumoniae infection and its treatment was conducted, with the findings critically compared and discussed in relation to our own results. Due to the absence of recent national data from Romania, European data were utilized as the primary benchmark for comparison.
Furthermore, the alignment of our study’s treatment strategies with current European guidelines was analyzed to assess consistency and relevance.
Round 2
Reviewer 1 Report
Comments and Suggestions for Authors
All of the comments raised during the second round of revision are listed in detail in the attached PDF file.

Author Response
Thank you very much for taking the time to review this manuscript. Please find the detailed responses below and the corresponding revisions/corrections highlighted/in track changes in the re-submitted files.
- The scientific name of the pathogen (Mycoplasma pneumoniae) should be kept in an italic style: Consider applying this notion to the whole manuscript.
All proper names of bacteria have been changed in the text, they are italicized in this version
- Regarding the chart title (Figure 1, Page 4): Consider adding the term (the) to read: “Incidence of M. pneumoniae cases over the past two decades at the National Institute of Infectious Diseases”.
All table and figure titles have been changed
- Line 162: The authors have thankfully indicated that the lowest incidence was recorded during the SARS-CoV-2 pandemic: Consider adding the exact timing of the era, to make the text more informative.
The pandemic period was mentioned accurately
- Since Table 1 has been revised to include the residence parameter, I recommend changing the title to read: “Demographic characteristics of the study participants.
All table and figure titles have been changed
- In Tables 1,2 ,3 and 4: The authors have added the terms (Fisher test p- value) to the title of the tables: I find this quite confusing and irrelevant. The authors have already indicated the statistical analysis used in the methodology section. Therefore, in case of significant data the following term (* Significant (P
All table and figure titles have been changed
- Figure 2: Page 7: The figure is poorly designed. Texts overlap with one another, which makes the chart look cluttered. To make the visual look neater, I recommend: • Adjusting the size of the text. • Adjusting the style of the text. • Elements should be properly spaced and adequately aligned. • Using consistent text format and style whenever possible.
The figure has been modified, the spaces have been enlarged for better visibility.
7Both tables and figures should be understandable as standalone graphics, therefore, the titles and content should be clear and informative.
All table and figure titles have been changed
- Titles of tables 2, 3, 4, and 5 are neither clear nor informative enough. • Consider adding clear descriptive titles to inform the audience about the exact purpose of each table.
I tried to describe the information in the table in more detail.
- The previous notion also applies to the legends of the newly-added figures (3 and 4). Kindly elaborate on describing the purpose of each figure.
I tried to describe the information in the table in more detail
Best regards,
MD.PHD Madalina Merisescu